# Idiopathic epiretinal membrane area changes in eyes with good vision and the association with visual function

**Su Hwan Park**[1], **Sangwoo Moon**[1,2], **Su-Jin Kim**[1,2], **Ji-Eun Lee**[1,2], **Sung Who Park**[2,3], **Iksoo Byon**[2,3], **Seung Min Lee**[1,2]*

**1** Department of Ophthalmology, Research Institute for Convergence of Biomedical Science and Technology, Pusan National University Yangsan Hospital, Yangsan, Gyeongsangnamdo, South Korea, **2** Pusan National University School of Medicine, Yangsan, Gyeongsangnamdo, South Korea, **3** Department of Ophthalmology, Medical Research Institute, Pusan National University Hospital, Busan, South Korea

* platinummetal@hanmail.net, macula@pusan.ac.kr

## Abstract

### Purpose

To evaluate idiopathic epiretinal membrane (iERM) area changes and the association with visual function in patients with good vision, using confocal scanning laser ophthalmoscopy (cSLO).

### Methods

The medical records of patients with iERM with best-corrected visual acuity (BCVA) ≥ 20/25 and a follow-up period ≥ 2 years were reviewed retrospectively. BCVA, metamorphopsia, fundus images using cSLO, and optical coherence tomography images were obtained at baseline and every 6 months. The ERM area and diagonal lengths of the long and short axes were determined. The progression group was defined as eyes with ≥2 lines of loss, and the stable group was defined as eyes with less change.

### Results

This study included 46 eyes, with 23 eyes in each group. Participants in the progression group were younger (60.7 vs. 65.7 years, P = 0.015) and showed a larger BCVA change (0.20 vs. 0.04, P < 0.001) and greater ERM area decrease (34.2% vs. 11.7%, P < 0.001) during the follow-up period. In the progression group, BCVA reduction was correlated with area decrease (β = 0.571, P = 0.010). No factors were associated with metamorphopsia. In the univariate analysis, ERM area reduction at 6 months was a significant predictor of vision loss (odds ratio, 1.632; P = 0.002). Receiver operating characteristic analysis identified a 7.66% ERM area reduction at 6 months as the optimal cutoff for predicting vision loss (81.8% sensitivity, 82.6% specificity).

**Data availability statement:** All relevant data are within the manuscript and its Supporting Information files.

**Funding:** The author(s) received no specific funding for this work.

**Competing interests:** The authors have declared that no competing interests exist.

## Conclusions

ERM area reduction significantly correlates with vision loss in patients with iERM with good vision. Monitoring ERM area changes can help predict disease progression and visual outcomes.

## Introduction

Idiopathic epiretinal membrane (ERM) is a condition where a fibrocellular membrane forms on the inner surface of the retina, leading to visual disturbances and metamorphopsia. The prevalence of the ERM in adults ranges from 3.4% to 8.8% and increases with age [1–7]. The ERM often remains stable but can progress in the 29–43% range [1,2,4,8]. Gradual visual function decline, including vision loss, metamorphopsia, and monocular diplopia, has led nearly 30% of patients to undergo surgery within 2 years [9].

Previous studies have investigated the natural course and progression of the ERM using various methods. Some studies have analyzed retinal thickness or attachment types using optical coherence tomography (OCT) [10,11], while others have evaluated parameters related to ERM contraction using fundus photography [1,2,8,12]. However, these studies mainly focused on investigating indirect factors related to ERM contraction, and no study has measured changes or the natural course of the ERM area. This is largely due to the limitations of traditional imaging with standard cameras, which make it challenging to accurately measure the ERM area and track its changes over time.

Recently, high-resolution confocal scanning laser ophthalmoscopy (cSLO) was developed and is widely used in clinical practice [13]. Confocal systems allow the capture of reflected light through a pinhole, which suppresses any scattered or reflected light outside the focal plane that could potentially blur the image. This results in a sharp, high-contrast image of an object layer located within the focal plane [13]. By reducing light scatter and improving image clarity, confocal scanning technology allows for detailed visualization of the delicate structures of the ERM. This improvement in imaging technology has facilitated a more precise assessment of the ERM area and its changes.

Visual impairment caused by the ERM is known to be associated with ERM contraction [7]. Tracking changes in the ERM area may help to assess the rate of contraction progression, identify related anatomical changes, and evaluate factors affecting visual function and their prognostic value for visual outcomes. Therefore, this study aimed to quantify the ERM area and investigate how changes in the ERM area affect visual function in patients with good baseline vision.

## Methods

### Study subjects

The medical records of patients with idiopathic ERM who underwent fundus examination at Pusan National University Yangsan Hospital between January 1, 2021 and February 29, 2024 were retrospectively reviewed, and the patients were consecutively enrolled. This study was conducted in accordance with the tenets of the Declaration of Helsinki and was conducted under the approval of the Institutional Review

Board (IRB) of Pusan National University Yangsan Hospital (No. 05-2023-072). The requirement for informed consent was waived by the IRB of the Pusan National University Yangsan Hospital owing to the retrospective design of this study and the use of anonymized data. Access for data collection was conducted from December 26, 2023 to November 30, 2024. No personal identification information other than the patient's medical record number was included during data collection, and all personal identification information was excluded before data processing.

The inclusion criteria for the study were as follows: (1) follow-up period of at least 2 years; (2) best-corrected visual acuity (BCVA) of 20/25 or better at baseline; and (3) ERM stage 2 or lower at baseline, as classified by Govetto et al. [14]. To minimize outcome variability due to the position and morphology of idiopathic ERM, only cases with thin fibrous membranes covering the central macula at baseline were included, while pseudoholes and lamellar holes were excluded.

Exclusion criteria were as follows: (1) eyes with vitreomacular traction; (2) glaucoma; (3) high myopia, defined as a spherical equivalent of ≥ −6.0 diopters or an axial length of ≥ 26 mm; (4) media opacities that hindered the evaluation of the fundus and OCT images, or cataracts graded ≥ 3 for nuclear opalescence or cortical cataracts according to the Lens Opacities Classification System III (LOCS III) [15]; (5) spontaneous resolution of the idiopathic ERM within the follow-up period; and (6) prior ocular surgery other than cataract surgery.

We investigated patient information such as age, sex, and follow-up period. Ocular parameters, ERM area, and retinal thickness were measured every 6 months and analyzed as follows.

## Ocular parameter measurements

At baseline and during each subsequent follow-up visit, the patients underwent a comprehensive ocular examination, including BCVA, metamorphopsia, aniseikonia, intraocular pressure (IOP), slit-lamp biomicroscopy, fundus photography, and OCT.

BCVA was measured using the Snellen visual acuity chart and converted to the logarithm of the minimum angle of resolution (logMAR). IOP was measured using a non-contact tonometer (Canon TX-20; Canon Inc., Tokyo, Japan). The spherical equivalent was obtained using an automated refractometer (RK-F2, Canon, Tokyo, Japan). The axial length was measured using partial coherence interferometry (IOL Master 500, Carl Zeiss Meditec AG, Jena, Germany). For all patients in this study, one retinal specialist (S.M.L or S.H.P) performed the slit-lamp fundus examination using a 90-diopter lens (Volk Optical, Mentor, USA). The lens status was determined by slit-lamp examination and graded according to the LOCS III system [15].

Metamorphopsia was assessed using the M-chart (Inami Co, Tokyo, Japan), which consists of 19 dotted lines with dot sizes of 0.1° and dot intervals ranging from 0.2° to 2.0° of visual angle [16]. The visual angle at which the patient perceived each dotted line as straight was recorded as the metamorphopsia score. Metamorphopsia scores for horizontal and vertical lines were individually measured.

Aniseikonia was evaluated using the Awaya New Aniseikonia Test (Handaya Co, Tokyo, Japan) [17]. Wearing red-green spectacles, the participants identified the pair of semicircles that appeared identical in size from plates with size differences increasing in 1% increments from 1% to 24%, measuring their level of aniseikonia. Aniseikonia measurements were not performed in cases with bilateral ERM.

## Image acquisition and measurement of the ERM area and lengths of axes

The area and axis lengths of the ERM were measured at baseline and at each follow-up visit using fundus photographs. Fundus images were acquired using the true color cSLO (Eidon AF™, CenterVue, Padova, Italy). This device uses a white LED light source (440–650 nm) to capture high-resolution images (14 megapixels, 4608 × 3288 pixels) with a 60° field of view. The fundus images were exported as quality-preserving JPEG files using a dedicated viewer.

The ERM area was measured using the ImageJ software (version 1.54; provided by the National Institutes of Health, Bethesda, MD, USA; https://imagej.nih.gov/ij/). A single investigator (S.H.P) manually traced the ERM boundaries on the

fundus image using a freehand selection tool and identified the boundaries by referencing the membrane reflection on the retinal surface in the fundus images. To ensure accuracy, both en face images and cross-sectional views from 7 x 7 mm 3D OCT scans were referenced to confirm and delineate the ERM area. A closed polygonal region of the ERM area was formed by tracing its boundary and was then added to the ROI (Region of Interest) Manager tools in ImageJ for pixel area measurement. Each pixel was determined to be 4.9 μm wide, and the area was calculated in mm$^2$ using the pixel-to-μm conversion ratio.

To further analyze sectional changes in the ERM and their effect on visual function, we divided the ERM area into four quadrants: superior, inferior, nasal, and temporal. The quadrants in the 5-mm circle were delineated by drawing a pair of perpendicular lines passing through the macular center using ImageJ. Next, we separately calculated the ERM area within each quadrant to examine regional variations over time. The macular center was determined by referencing the vertex of the foveal bulge on OCT.

Within the ERM area, we measured the following types of axes passing through the fovea: the lengths of the horizontal and vertical axes, and the lengths of the long and short axes.

In addition to the area and axial length of the ERM measured at each follow-up, the ratio of the reduction area and decreased axial length in the ERM at each follow-up was analyzed and compared with the values measured at baseline. The reduction area or decreased length was defined as the value measured at baseline minus the value measured at follow-up, while the reduction area ratio and decreased length ratio were defined as the difference divided by the baseline measurement value. To evaluate the shape of the ERM, the eccentricity at baseline and final follow-up were evaluated using two long and short axes according to the eccentricity formula as follows:

$$\text{Eccentricity} = \sqrt{1 - \left(\frac{short\ axis\ length}{long\ axis\ length}\right)^2}$$

## Measurement of the retinal thickness and photoreceptor layer

To investigate the relationship between changes in the ERM area and retinal thickness, we measured the mean total retinal thickness, mean inner retinal thickness, and mean outer retinal thickness. ERM stage was assessed using OCT at baseline and at each follow-up visit.

Retinal thickness was analyzed using swept-source OCT (Triton, Topcon Corp., Tokyo, Japan) and automatically determined using the embedded OCT software algorithm. To analyze regional changes in retinal thickness, an Early Treatment Diabetic Retinopathy Study (ETDRS) grid was used. The boundaries of the inner and outer retina were automatically divided using the embedded software, and the mean thickness of the inner and outer retina in each section was measured. Inner retinal thickness was defined as the distance between the internal limiting membrane and inner nuclear layer, while outer retinal thickness was defined as the distance between the outer plexiform layer and retinal pigment epithelium. The ETDRS grid divides the macula into three ring regions with four quadrants: central 1-mm ring (central subfield macular thickness, CSMT), inner 3-mm ring (comprising superior, nasal, inferior, and temporal quadrants), and outer 3-mm ring (comprising superior, nasal, inferior, and temporal quadrants).

The ERM was classified into four stages using the OCT classification system proposed by Govetto et al. [14]. Ectopic inner foveal layer (EIFL) was defined as a continuous and clearly identified inner nuclear layer (hyporeflective) and inner plexiform layer (hyperreflective) band crossing through the fovea and it was evaluated using an OCT-based staging scheme of ERM.

## Progression group and stable group

Significant vision loss was defined as the loss of ≥ 2 lines on the Snellen visual acuity chart. The participants were divided into two groups. The progression group included eyes with vision loss of ≥ 2 lines due to ERM progression within the follow-up period, whereas the stable group consisted of eyes without such vision loss.

 

## Surgical indication

During the follow-up period, pars plana vitrectomy was performed to remove the ERM if the BCVA decreased to less than 20/32 with visual symptoms, or if other visual functions such as metamorphopsia or diplopia deteriorated despite a BCVA of 20/32. Patients who underwent surgery within 12 months were excluded because of the difficulty in assessing the natural disease course. For patients who underwent surgery after 12 months, the last fundus photograph and the BCVA measurement obtained prior to surgery were used for analysis.

## Statistical analysis

The Statistical Package for the Social Sciences (SPSS) for Windows (version 23.0; SPSS Inc., Chicago, IL, USA) was used for statistical analysis. Normality tests were performed using the Shapiro-Wilk test. Data are presented as mean±standard deviation for variables with a normal distribution, and as median and interquartile range for variables with skewed distributions. Continuous variables were compared between groups using independent t-tests or Mann-Whitney U tests, depending on the distribution. Categorical variables were compared using Chi-square or Fisher's exact tests. Paired data were compared using either the paired sample t-test or Wilcoxon signed-rank test, as appropriate. The correlation between BCVA changes and ERM-related factors (metamorphopsia, aniseikonia, and area reduction or length decrease ratio of the ERM) was analyzed using Pearson's correlation coefficient.

Simple linear regression and stepwise multiple linear regression analyses were used to investigate the factors associated with BCVA changes. Variance inflation factor (VIF) was calculated to measure multicollinearity among variables. A VIF > 10 indicates significant multicollinearity.

Univariate analysis was performed for each variable to analyze factors related to significant vision loss in the ERM, and multivariate logistic regression analysis was performed for factors with a P-value < 0.1 in the univariate analysis. Statistical significance was defined as a P-value < 0.05.

Receiver operating characteristic (ROC) curves were used to evaluate the predictive accuracy of ERM area reduction for significant vision loss. This was measured using the area under the ROC curve (AUROC) and included sensitivity and specificity calculations for these parameters.

## Results

This study included 46 consecutive eyes of 43 patients diagnosed with idiopathic ERM who met the inclusion and exclusion criteria. Each group consisted of 23 eyes.

The baseline characteristics of patients in the progression and stable groups are summarized in Table 1.

Participants in the progression group (mean age, 60.7±6.6 years) were significantly younger than those in the stable group (mean age, 65.7±6.6 years) (P=0.015). No significant differences were observed between groups in terms of baseline status, sex, lens status, BCVA, IOP, axial length, CSMT, laterality, or follow-up period.

In the stable group, nuclear opalescence and cortical cataract scores were 2.0±0.0 and 0.9±0.7 at baseline, and 2.2±0.4 and 0.9±0.7 at the final follow-up, respectively. In the progression group, the scores were 1.9±0.2 and 0.6±0.7 at baseline, and 2.0±0.4 and 0.6±0.7 at the final follow-up, respectively. There was no significant progression of cataract in either group.

## Changes in BCVA, metamorphopsia, and aniseikonia

Comparisons of changes in BCVA, metamorphopsia, and aniseikonia between the two groups from baseline to the final follow-up are presented in Table 2.

No significant differences in metamorphopsia and aniseikonia were observed between the groups during follow-up; however, the BCVA in the progression group was significantly worse than that at baseline from 12 months to the final follow-up (P=0.010, 0.007, and 0.001, respectively). The change in BCVA from baseline was significantly greater in the progression group than that in the stable group, since the 6-month follow-up (Fig 1).

**Table 1. Baseline characteristics of patients.**

| Variable | Progression group | Stable group | P-value |
|---|---|---|---|
| Eyes (number) | 23 | 23 | – |
| Age (years) | 60.7 ± 6.6 | 65.7 ± 6.6 | 0.015 |
| Sex (male/ female) | 12/ 11 | 13/ 10 | 1.000* |
| Follow-up period (months) | 26.9 ± 6.4 | 27.4 ± 7.1 | 0.818‡ |
| BCVA (logMAR) | 0.02 ± 0.09 | 0.05 ± 0.07 | 0.220‡ |
| IOP (mmHg) | 16.6 ± 3.1 | 16.0 ± 2.4 | 0.463 |
| Spherical equivalent (diopter) | 0.50 ± 1.44 | 0.31 ± 1.55 | 0.956‡ |
| Axial length (um) | 23.33 ± 0.59 | 23.10 ± 1.11 | 0.658 |
| CSMT (um) | 373.0 ± 56.4 | 360.5 ± 52.9 | 0.442 |
| OCT stage | 1.7 ± 0.4 | 1.6 ± 0.5 | 0.351‡ |
| Laterality (Right/Left) | 12/ 11 | 13/ 10 | 0.767* |
| Lens status (Phakic/Pseudophakic) | 22/ 1 | 20/ 3 | 0.608† |
| Metamorphopsia Horizontal Score | 0.24 ± 0.34 | 0.12 ± 0.19 | 0.358‡ |
| Metamorphopsia Vertical Score | 0.24 ± 0.37 | 0.14 ± 0.24 | 0.461‡ |
| Aniseikonia (%) | 3.5 ± 5.0 | 2.4 ± 2.4 | 0.820‡ |

Values are presented as mean ± standard deviation, median (interquartile range), or number unless otherwise indicated.

BCVA, best-corrected visual acuity; CSMT, central subfield macular thickness; IOP, intraocular pressure; logMAR, logarithm of the minimum angle of resolution; OCT, optical coherence tomography.

Comparison between groups was performed using the independent t-tests or the Mann–Whitney U test.

* Pearson's Chi-squared test results.

† Fisher's exact test results.

‡ Mann Whitney U-test results.

**Table 2. Comparison of ERM areas and lengths between the two groups at the final follow-up.**

| Variable | Progression group | Stable group | P-value |
|---|---|---|---|
| BCVA (logMAR) | 0.22 ± 0.10 | 0.09 ± 0.09 | <0.001 |
| Metamorphopsia Horizontal Score | 0.47 ± 0.56 | 0.34 ± 0.43 | 0.497 |
| Metamorphopsia Vertical Score | 0.37 ± 0.44 | 0.34 ± 0.39 | 0.850 |
| Aniseikonia (%) | 4.1 ± 4.3 | 4.2 ± 4.2 | 0.810 |
| ΔBCVA (logMAR) | 0.20 ± 0.00 | 0.04 ± 0.05 | <0.001 |
| ΔMetamorphopsia Horizontal Score | 0.26 ± 0.60 | 0.22 ± 0.36 | 0.820 |
| ΔMetamorphopsia Vertical Score | 0.14 ± 0.35 | 0.20 ± 0.23 | 0.593 |
| ΔAniseikonia (%) | 0.80 ± 4.54 | 2.17 ± 4.22 | 0.123 |

Values are presented as mean ± standard deviation.

BCVA, best-corrected visual acuity; ERM, Epiretinal membrane; logMAR, logarithm of the minimum angle of resolution.

Δ = value at final follow-up - value at baseline

Comparison between groups was performed using the Mann–Whitney U test.

At the final follow-up, the BCVA change in the progression group was 0.20 ± 0.00, whereas that in the stable group was 0.04 ± 0.05 (P < 0.001). In the progression group, the change in BCVA was not correlated with changes in metamorphopsia or aniseikonia; however, the change in vertical metamorphopsia was correlated with the change in aniseikonia (horizontal: rho ($\rho$)=0.497, P = 0.100; vertical: $\rho$ = 0.721, P = 0.008). In the stable group, none of these three factors were correlated. During the follow-up period, ERM surgery was performed in 9 out of 46 eyes (19.6%) with a median BCVA of 0.30

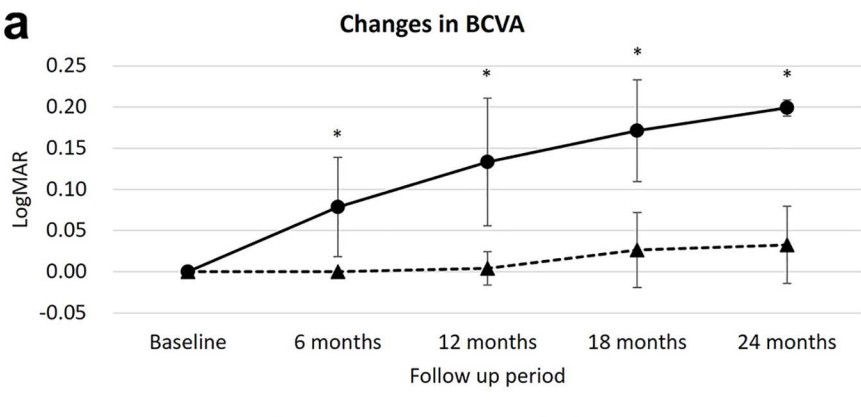

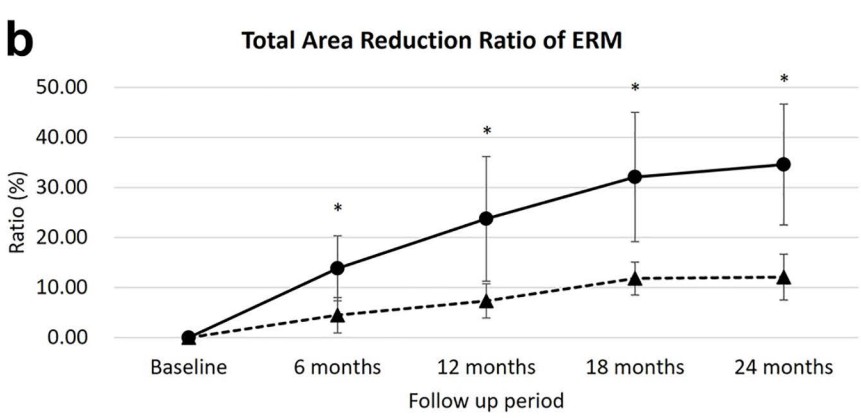

| | Baseline | 6 months | 12 months | 18 months | 24 months |
|---|---|---|---|---|---|
| Progression group (N) | 23 | 22 | 23 | 19 | 19 |
| Stable group (N) | 23 | 23 | 23 | 19 | 23 |

**Fig 1. Changes in best-corrected visual acuity and total area reduction ratio of the epiretinal membrane (ERM) during follow-up.** (a) During the follow-up period, the stable group showed no significant change in visual acuity, whereas the progression group exhibited a decrease compared to base-line starting at 12 months, with greater changes than those in the stable group beginning at 6 months. (b) The total ERM area decreased from baseline at 6 months in both groups; however, the total area reduction ratio of the ERM was significantly greater in the progression group from 6 months onward. *Statistically significant difference (P<0.05) between two groups.

(Snellen acuity 20/40, interquartile range: 0.30–0.30), all of which belonged to the progression group. Four of these eyes underwent surgery within 24 months.

### Changes in the ERM area and length

Comparisons of the ERM area and length, reduction areas of ERM, decreased lengths, and associated ratios between the two groups at baseline and final follow-up are presented in Table 3. Fig 2 shows ERM area and length measurements as well as ERM changes in representative cases of the progression and stable groups.

**Table 3. Comparison of changes in ERM areas and lengths between the two groups at baseline and final follow-up.**

| Variable | Progression group | Stable group | P-value |
|---|---|---|---|
| Area of ERM at baseline | | | |
| Total area (mm2) | 16.42±7.58 | 15.26±7.80 | 0.613 |
| Superior area (mm2) | 4.61±3.64 | 4.61±2.83 | 0.993 |
| Inferior area (mm2) | 3.36±2.27 | 3.02±2.74 | 0.642 |
| Nasal area (mm2) | 3.26±2.34 | 3.13±2.57 | 0.865 |
| Temporal area (mm2) | 5.21±2.64 | 4.53±2.39 | 0.367 |
| Length of ERM at baseline | | | |
| Horizontal length (mm) | 4.37±1.07 | 3.79±1.13 | 0.081 |
| Vertical Length (mm) | 3.85±1.34 | 3.86±1.49 | 0.981 |
| Long Length (mm) | 5.54±1.49 | 5.21±1.73 | 0.488 |
| Short length (mm) | 3.12±0.88 | 3.07±1.00 | 0.848 |
| Reduction Area of ERM | | | |
| Total area ($mm^2$) | 5.72±3.47 | 1.78±1.10 | <0.001 |
| Superior area ($mm^2$) | 2.09±2.84 | 0.23±0.87 | 0.001 |
| Inferior area ($mm^2$) | 1.00±0.93 | 0.56±0.65 | 0.047 |
| Nasal area ($mm^2$) | 1.04±0.93 | 0.42±0.65 | 0.018 |
| Temporal area ($mm^2$) | 1.58±−0.56 | 1.29±0.53 | 0.001 |
| Decreased Length of ERM | | | |
| Horizontal length (mm) | 0.93±0.78 | 0.18±0.33 | <0.001 |
| Vertical Length (mm) | 1.03±0.92 | 0.17±0.39 | <0.001 |
| Long Length (mm) | 0.87±0.73 | 0.18±0.33 | <0.001 |
| Short length (mm) | 0.79±0.64 | 0.31±0.28 | 0.010 |
| Area Reduction Ratio of ERM | | | |
| Total area (%) | 34.2±11.6 | 11.7±4.6 | <0.001 |
| Superior area (%) | 36.6±22.6 | 7.4±21.5 | <0.001 |
| Inferior area (%) | 32.0±27.2 | 13.0±24.8 | 0.018 |
| Nasal area (%) | 36.1±28.2 | 6.9±68.4 | 0.001 |
| Temporal area (%) | 29.9±20.7 | 13.5±12.6 | 0.001 |
| Length Decrease Ratio of ERM | | | |
| Horizontal length (%) | 21.6±17.6 | 3.5±9.1 | <0.001 |
| Vertical Length (%) | 25.5±18.2 | 3.8±11.0 | <0.001 |
| Long Length (%) | 14.7±11.2 | 4.1±7.1 | <0.001 |
| Short length (%) | 25.1±17.1 | 8.4±10.1 | <0.001 |

Area Reduction ($mm^2$) = Area at Final follow-up – Area at Baseline

Decreased Length (mm) = Length at Final follow-up – Length at Baseline

Area Reduction Ratio (%) = Area Reduction/ Baseline Area ×100

Length Decrease Ratio (%) = Decreased Length/ Baseline Length ×100

At baseline, no statistically significant differences were observed in the ERM area, including the total area and each quadrant (superior, inferior, nasal, and temporal), or in axial lengths (horizontal, vertical, long, and short axes) between the two groups. In both groups, a significant decrease in the total ERM area was observed from 6 months (P < 0.001, Fig 1), and each quadrant showed a significant decrease, except for the upper quadrant of the stable group (P < 0.05). Except for the length of the long axis in the stable group, the lengths of the other axes significantly decreased from 6 months (P < 0.05). However, when comparing the two groups, the ERM area changes (total and all

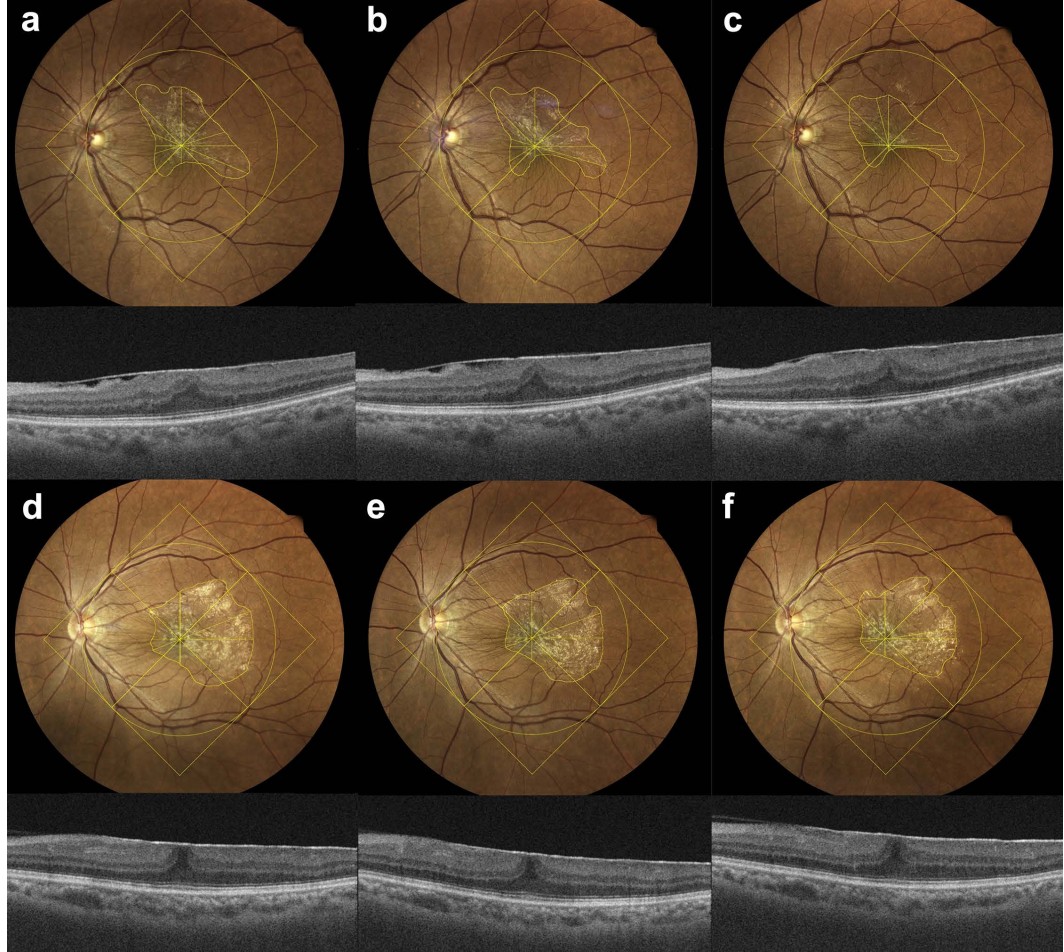

**Fig 2. Representative cases showing changes in the epiretinal membrane (ERM) area.** In the progression group, fundus photographs (upper panel) show progressive contraction of the ERM area from baseline (a) to 12 months (b) and 24 months follow-up (c). ERM contraction occurred predominantly along the short axis, resulting in a more eccentric elliptical shape over time. This contraction was accompanied by an increase in retinal thickness on ocular coherence tomography (lower panel). The stable group showed little change in the ERM area from baseline (d) to 12 months (e) and 24 months (f) (upper panel). Retinal thickness remained relatively unchanged (lower panel).

quadrants) were significantly greater in the progression group from 6 months (P < 0.05, Fig 1), and the axial length changes (horizontal, vertical, long, and short axes) were significantly greater in the progression group from 12 months (P < 0.05) (Table 3).

In Addition, the ratio of ERM area reduction was significantly different between the groups from 6 months, with the progression group having a total area reduction of 13.9 ± 6.5%, 23.7 ± 12.5%, 32.1 ± 11.6%, and 34.2 ± 11.6%, compared to 4.5 ± 3.6%, 7.4 ± 3.4%, 11.4 ± 3.6%, and 11.7 ± 4.6% in the stable group at 6, 12, 18, and 24 months, respectively (P < 0.001). In each group, no difference was observed in the area reduction ratio between quadrants. The median change in BCVA according to the area reduction ratio was 0.0 (0.0–0.1; 8 eyes) at <10%, 0.0 (0.0–0.1; 15 eyes) at 10–20%, and 0.2 (0.2–0.2; 23 eyes) at ≥20%. The ratio of length decrease in all axes was greater in the progression group than that in the stable group at 12 months follow-up (P < 0.05). In terms of eccentricity, no significant change was observed in the stable group (from 0.78 ± 0.10 to 0.79 ± 0.14), whereas a significant increase was observed in the progression group from 0.79 ± 0.10 to 0.84 ± 0.10 (P = 0.05).

## Changes in the ERM area and retinal thickness

Regarding the total, inner, and outer retinal thickness measured by OCT, evaluations across all ETDRS grid regions showed a significant increase in retinal thickness in both the central 1-mm ring and inner 3-mm ring, except for inner retinal thickness in the inner 3-mm ring of the stable group, which was observed in both groups at the final follow-up (P<0.05 for all; S1 Table). No significant differences were observed in the total retinal thickness, inner retinal thickness, or outer retinal thickness in the central 1-mm ring, inner 3-mm ring, or outer 6-mm ring between the two groups at baseline and final follow-up (S1 Table). Thickness changes in the total, inner, and outer retina also showed no significant differences between both groups. In addition, thickness changes showed no correlations with visual function, including BCVA, metamorphopsia, and aniseikonia.

When comparing the thickness change according to location, the increase in retinal thickness in the progression group was greater in the central 1-mm ring than that in the inner 3-mm ring, and greater in the inner 3-mm ring than that in the outer 6-mm ring (P=0.06 and 0.010, respectively). No significant difference in retinal thickness change was observed in each quadrant in the progression group or in these locational factors in the stable group.

Regarding the association between the ERM area reduction ratio and retinal thickness change, in the progression group, area reduction in the temporal quadrant was associated with increased retinal thickness in the central 1-mm and inner 3-mm rings (P<0.05, S2 Table).

In the stable group, no eyes showed a change in OCT stage during the follow-up period. However, in the progression group, the OCT stage significantly increased from a mean of 1.7±0.4 at baseline to 1.9±0.5 at the final follow-up (P=0.014). Specifically, stage progression was observed in 3 eyes from stage 1–2 and in another 3 eyes from stage 2–3.

## Correlation between changes in the ERM area and visual function

BCVA changes correlated with the reduction ratios of the total, superior, and temporal areas of the ERM, as well as the decreased ratios of the long and short lengths (P<0.05 for all; Table 4).

On multiple linear regression analysis, BCVA changes were only significantly associated with the total area reduction ratio of the ERM (β=0.571, P=0.010). No significant association was observed between BCVA and the other factors

**Table 4. Simple and multiple linear regression analyses between clinical factors and visual acuity change at final follow-up.**

| Variable | Pearson's Correlation | | Simple liner | | | | Multiple | | |
|---|---|---|---|---|---|---|---|---|---|
| ΔBCVA (LogMAR) | R correlation coefficient | P-value | Coeffi-cient B | Standard Regres-sion Coefficient (β) | Standard error | P-value | Standard Regres-sion Coefficient (β) | Standard error | P-value |
| ΔMetamorphopsia Horizontal Score | −0.094 | 0.635 | −0.013 | −0.094 | 0.027 | 0.635 | | | |
| ΔMetamorphopsia Vertical Score | −0.152 | 0.441 | −0.034 | −0.152 | 0.044 | 0.441 | | | |
| ΔAniseikonia (%) | −0.376 | 0.070 | −0.006 | −0.275 | 0.004 | 0.194 | | | |
| Area Reduction Ratio of ERM | | | | | | | | | |
| Total area (%) | 0.662 | <0.001 | 0.003 | 0.662 | 0.001 | <0.001 | 0.571 | 0.001 | 0.010 |
| Superior area (%) | 0.506 | <0.001 | 0.001 | 0.506 | 0.000 | <0.001 | 0.207 | 0.000 | 0.167 |
| Inferior area (%) | 0.233 | 0.119 | 0.001 | 0.233 | 0.000 | 0.119 | | | |
| Nasal area (%) | 0.206 | 0.170 | 0.000 | 0.206 | 0.000 | 0.170 | | | |
| Temporal area (%) | 0.450 | 0.002 | 0.002 | 0.450 | 0.001 | 0.002 | 0.214 | 0.000 | 0.097 |
| Length Decrease Ratio of ERM | | | | | | | | | |
| Long Length (%) | 0.430 | 0.003 | 0.003 | 0.430 | 0.001 | 0.003 | −0.066 | 0.001 | 0.669 |
| Short length (%) | 0.353 | 0.016 | 0.002 | 0.353 | 0.001 | 0.016 | −0.131 | 0.001 | 0.392 |

BCVA, best-corrected visual acuity; ERM, epiretinal membrane

Δ = value at final follow-up – value at baseline

associated with ERM changes. Multicollinearity was not observed among the independent variables (VIF < 5). No factors were correlated with metamorphopsia or aniseikonia.

## Changes in the ERM area and predictive factors

A univariate logistic regression analysis was conducted using the total area of the ERM at baseline and total area reduction ratio of the ERM at 6, 12, and 18 months to identify factors associated with significant vision loss of ≥ 2 lines in the overall group (Table 5).

The ERM area reduction ratio at 6, 12, and 18 months was found to be a significant factor influencing vision loss (P = 0.002, 0.003, and 0.028, respectively). However, the total ERM area at baseline was not significantly different.

ROC curve analysis was performed to evaluate the predictive accuracy of the total ERM area reduction ratio, from baseline to 6 months, in distinguishing significant vision loss in the overall group (Fig 3).

The analysis revealed that the AUROC of the total area reduction ratio of the ERM over the first 6 months was 0.911 (P < 0.001), with an optimal cutoff value of 7.66%, resulting in 81.8% sensitivity and 82.6% specificity for identifying significant vision loss (S3 Table).

## Discussion

Idiopathic ERM primarily develops due to posterior vitreous detachment or abnormal vitreoretinal separation caused by vitreous liquefaction [18,19]. During this process, Müller cells, hyalocytes, and retinal pigment epithelial cells may undergo transdifferentiation into myofibroblasts [7,18]. These transformed cells exhibit proliferative and contractile properties and can produce extracellular matrix, leading to thickening and contraction of the ERM [7,20,21]. The membrane exerts a tangential traction force on the underlying retinal tissue, damaging retinal function and partially obscuring the macula, causing visual impairment [7].

Studies on the natural history of ERM progression over time have reported that approximately 29–43% of cases show progression [1,2,4,8]. In attempts to assess these changes, fundus photography has been used to evaluate changes in membrane area or to measure distances between specific anatomical landmarks [1,8,12,22,23]. However, measuring the area of the ERM using conventional fundus photography has limitations in that clear identification of the ERM border is difficult due to technical constraints. The method using specific retinal landmarks—including measuring the distances between vascular bifurcation points, analyzing regional vector displacements, calculating total changes in distances between the optic disc and vascular arcades, and observing angular changes between retinal structures intersecting a reference circle—has allowed for a more reliable evaluation of ERM contraction and progression over time, although it does not directly reflect changes in the ERM itself [8,12,22,23].

Most recent studies have identified ERM changes using OCT rather than fundus imaging because of the limitations of conventional fundus photography [14,24–28]. To evaluate anatomical changes related to ERM, some studies have used cross-sectional analysis, while others have employed en face imaging or OCT angiography [14,24–28]. However, these methods using OCT are unlikely to represent changes in the overall shape of the ERM and may not match well with the

**Table 5. Univariate analysis of the total ERM area in relation to significant vision loss.**

| Variable | OR | 95% CI | P-value |
|---|---|---|---|
| Total Area of ERM at baseline | 1.021 | 0.945–1.103 | 0.605 |
| Total Area Reduction Ratio of ERM after 6 months | 1.632 | 1.205–2.210 | 0.002 |
| Total Area Reduction Ratio of ERM after 12 months | 1.327 | 1.099–1.602 | 0.003 |
| Total Area Reduction Ratio of ERM after 18 months | 1.820 | 1.067–3.103 | 0.028 |

ERM, epiretinal membrane

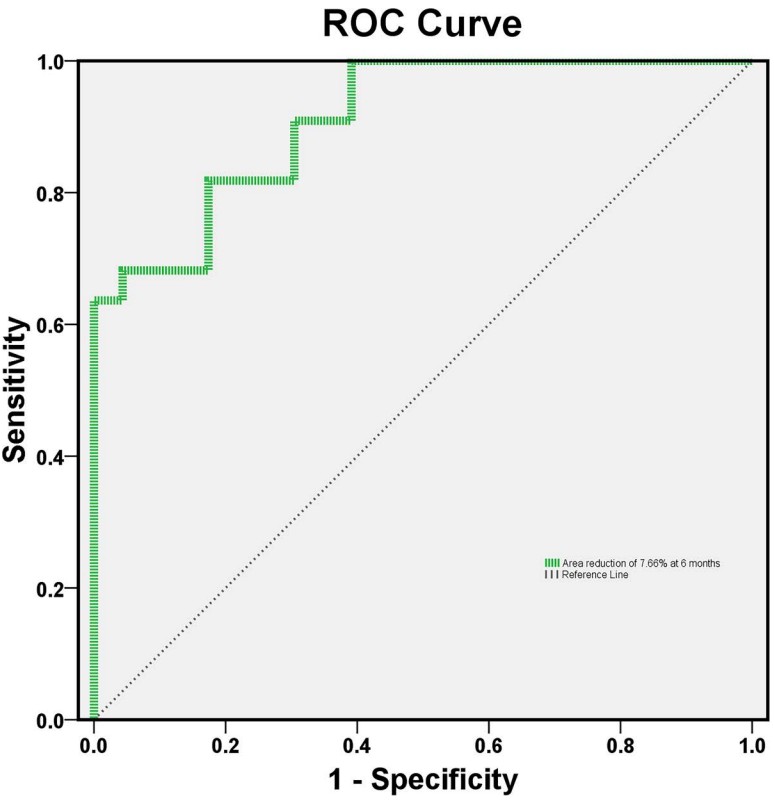

**Fig 3. Receiver operating characteristic curve for predicting 24-month vision loss using the total area reduction ratio at 6-month follow-up.**
A total area reduction ratio of 7.66% at the 6-month follow-up predicted a ≥ 2-line visual acuity loss at 24 months with 81.8% sensitivity and 82.6% specificity.

shape of the ETDRS ring, which measures the average thickness. Variations based on measurement site may exist, which may pose limitations in detecting early changes or performing quantitative correlation analyses.

Therefore, evaluating the ERM area may be useful to quantify the degree of change and assess its impact on visual function. Compared to traditional fundus photography, cSLO-based fundus imaging allows for clearer visualization of the ERM boundaries, enabling accurate measurement of its area and detection of changes over time [13,29–31]. Additionally, a study has shown that cSLO can better visualize the ERM than en face OCT or pseudocolor images [32]. cSLO imaging generally offers a broader field of view (approximately 30–60°) compared to en face imaging, enabling rapid visual evaluation and suggesting higher clinical usefulness.

In this study, the ERM area was measured using images captured by cSLO, and its relationship with visual function was investigated. Consequently, we confirmed that a decrease in the ERM area over time was related to visual acuity loss in the progression group. Patients in the progression group were younger than those in the stable group. ERM area reduction was identified after 6 months of follow-up in both groups; however, the amount of change in the area and axial length was greater in the progression group than that in the stable group. The ERM shape changed more elliptically in the progression group than in the stable group. In both groups, retinal thickness increased, especially in the 1-mm ring; however, the difference between the two groups was not statistically significant. Among the various factors related to the area or axial length, the total area reduction ratio of the ERM was the most important factor influencing changes in visual acuity. An area change of 7.66% at 6 months follow-up predicted a two-line loss in visual acuity within 24 months, with a

sensitivity of 81.8% and a specificity of 82.6%. The factors investigated in the present study did not correlate with metamorphopsia or aniseikonia.

During the 24-month follow-up, 50% of the eyes experienced a decrease in visual acuity of ≥ 2 lines. These findings are comparable to those of Kofod et al., who reported that 24% of patients with good baseline visual acuity experienced sufficient deterioration to require surgery within 12 months [33]. However, the average decline in visual acuity observed in our study was greater than that reported by Luu et al., who found a decrease of 0.012±0.003 logMAR units per year [34]. This difference may be attributed to factors such as the younger age of patients in our study, with a mean age of 60.7 years in the progression group and 65.7 years in the stable group, compared to 70.6 years in the study by Luu et al. Likewise is the better baseline visual acuity in our cohort, which ranged from 0.02 to 0.05, compared to 0.17 in Luu et al.'s study [34].

Participants in the progression group were younger than those in the stable group. Although ERM prevalence increases with age, the greater progression observed in younger patients may be attributed to factors such as reduced transdifferentiation into myofibroblasts, decreased contractile activity, or lower expression of growth factors and cytokines associated with aging [35–38]. Clinically, Chen et al. showed that the rate of progression to surgery did not increase beyond 4 years of follow-up over a total follow-up period of 7 years [39]. Similarly, in our study, ERM area changes measured at 6-month intervals tended to decrease as follow-up approached 24 months. These findings suggest that progression mostly occurs early in the course of ERM or that the disease becomes less active with age, resulting in fewer changes over time.

Regarding metamorphopsia and aniseikonia, no significant changes were found in both groups. Although the exact reason remains unclear, the lack of correlation between visual acuity and metamorphopsia in the ERM has been consistently reported in several previous studies [40,41]. In our two previous studies, parameters based on the distance between the optic disc and the vascular arcade, representing a wide area change, showed an association with visual acuity changes, but failed to demonstrate a correlation with metamorphopsia [12,42]. Studies investigating the association between metamorphopsia and anatomical features have identified localized changes near the fovea, such as altered perifoveal vessel spacing, changes in the foveal avascular zone, and inner nuclear layer thickness on OCT, as relevant factors [28,43,44]. These findings suggest that the present study, which focused on the overall area of the ERM and the thickness of the total retina or its inner and outer layers, may be inadequate to explain the observed changes in metamorphopsia.

Although the ERM area decreased in both groups, the reduction was greater in the progression group. Owing to the lack of previous studies that have specifically evaluated changes in the ERM area, direct comparisons with our results are limited. However, our findings appear to align with those of Kofod et al., who reported that eyes with greater vector displacement of the retinal vessels had poorer visual acuity than their fellow eyes [22]. Similarly, in our previous study, eyes in the progression group, defined by a loss of more than two lines of visual acuity over 24 months, showed a greater reduction in the sum of distances between the disc, fovea, and vessels by approximately 1.8%, compared to only 0.5% in the control group [12].

The lengths of both long and short axes decreased after surgery in both groups. In the stable group, the changes in axis length were not significantly different. In the progression group, the decrease in the length of the short axis was greater than that of the long axis. Consequently, eccentricity, which represents the degree of ellipticity, increased in the progression group, indicating that the overall shape of the ERM became more elliptical.

Regarding changes in retinal thickness, the progression group showed a particularly notable increase in the inner retina of the central 1-mm ring. Although previous studies have reported that thickening tends to occur more prominently in the inner retina in eyes with ERMs, our study further demonstrated that the central 1-mm ring exhibited greater thickening than the surrounding areas [25,45]. These findings may be related to centripetal contraction of the ERM, traction on the inner retina leading to earlier structural changes, and the relatively small area of the central region. However, because our study focused on early changes in visual acuity, many patients were in the initial stages of the disease, which may have limited the development of significant differences in retinal thickness between the two groups. Unlike previous studies,

changes in ERM thickness did not show a significant difference between the progression and stable groups, and no clear correlation between thickness changes and visual function were observed.

The area change in the temporal quadrant of the progression group was associated with total retinal thickness changes in both the 1-mm and 3-mm regions. This may be due to the relatively greater mobility of the temporal retina than that of other directions, possibly influenced by the physical barrier effect of the optic nerve head, resistance to deformation by the vascular arcade, and a higher density of retinal nerve fiber layers [46–48]. In other studies, postoperative findings showed nasal displacement of the retina, a decrease in the foveal angle, and a shift of the fovea toward the optic disc [23,42,49]. These changes may have occurred because the retina had already contracted and was displaced temporally during the natural course before surgery, as observed in our study.

Among the factors affecting visual acuity, the total area reduction ratio of the ERM showed the strongest association in the multivariate regression analysis. This area change ratio was consistently correlated with visual acuity decline across all follow-up time points based on regression analysis. Notably, a reduction in the ERM area of 7.66% or more at 6 months, a clinically relevant time point for predicting disease progression, predicted visual acuity loss of two or more lines at 24 months. These findings suggest that unlike other ERM-related parameters that are difficult to quantify or not easily applicable in clinical practice, the total area reduction ratio can be measured objectively and may serve as a clinically useful predictor of visual decline.

Our study has several limitations. First, the relatively small sample size and retrospective design may limit the generalizability of our findings. Second, the relatively short follow-up duration may have influenced the observed progression rates and visual outcomes. Third, although eyes with cataract that interfered with visual acuity were excluded from the study, mild cataract progression was noted in our samples, which may have influenced visual impairment in eyes with membrane progression. Nonetheless, previous research suggests that nuclear opalescence and cortical cataracts of grade 2 or less have a negligible effect on vision after surgery [50,51]. Fourth, manual measurement of the ERM area may introduce observer bias and variability. Furthermore, the potential effects of image rotation or misalignment on fundus photography could further affect measurement accuracy. Fifth, because the criteria for the progression group were based on a visual acuity loss of two or more lines, patients in this group easily reached the surgical threshold, and the margin between baseline and preoperative visual acuity was narrow. Consequently, the mean visual acuity loss in the progression group was 2.0 lines with a standard deviation of 0. This uniformity in visual acuity may limit assessment of the correlation between varying degrees of visual decline and ERM area changes. Finally, although the ERM may appear transparent in the early stages and gradually become opaque or increase in area due to fibrocellular proliferation in some cases, these aspects were not considered in this study. However, we believe that this limitation did not significantly affect our results because no cases in our dataset showed area enlargement at 24 months. Future studies with larger cohorts and standardized follow-up protocols are required to validate our findings and provide a more comprehensive understanding of ERM progression. Nevertheless, our study seems to be the first to use fundus photography to provide new information with respect to the natural course of ERM area changes in eyes with good vision.

## Conclusions

This study enhances our understanding of ERM area changes and the relationship with visual function. We found that a reduction in the ERM area was strongly correlated with vision loss. Monitoring the decrease in the total area of the ERM provides valuable insights into disease progression and its impact on visual acuity. Notably, changes in the ERM area within the first 6 months are particularly critical, offering important indicators of disease advancement and potential visual outcomes.

## Supporting information

**S1 File. Data.**
(XLSX)

**S1 Table. Comparison of retinal thickness changes between the two groups at baseline and final follow-up.**
(DOCX)

**S2 Table. Correlation between ERM area changes and retinal thickness changes.**
(DOCX)

**S3 Table. The values of sensitivity, specificity, and AUROC for each ratio.**
(DOCX)

## Author contributions

**Conceptualization:** Su-Jin Kim, Ji-Eun Lee, Seung Min Lee.

**Data curation:** Su Hwan Park, Seung Min Lee.

**Formal analysis:** Su Hwan Park, Seung Min Lee.

**Investigation:** Su Hwan Park, Seung Min Lee.

**Methodology:** Su Hwan Park, Sangwoo Moon, Su-Jin Kim, Ji-Eun Lee, Iksoo Byon, Seung Min Lee.

**Project administration:** Seung Min Lee.

**Supervision:** Seung Min Lee.

**Validation:** Sangwoo Moon, Su-Jin Kim, Ji-Eun Lee, Iksoo Byon.

**Visualization:** Seung Min Lee.

**Writing – review & editing:** Sangwoo Moon, Sung Who Park, Iksoo Byon, Seung Min Lee.

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
