## [Decision Letter · Decision Letter 0]

29 Jul 2025

PONE-D-25-36863Idiopathic Epiretinal Membrane Area Changes in Eyes with Good Vision and the Association with Visual FunctionPLOS ONE

Dear Dr. Lee,

Thank you for submitting your manuscript to PLOS ONE. After careful consideration, we feel that it has merit but does not fully meet PLOS ONE’s publication criteria as it currently stands. Therefore, we invite you to submit a revised version of the manuscript that addresses the points raised during the review process.

We look forward to receiving your revised manuscript.

Kind regards,

Daisuke Nagasato

Academic Editor

PLOS ONE

Journal Requirements:

Reviewers' comments:

Reviewer's Responses to Questions

**Comments to the Author**

1. Is the manuscript technically sound, and do the data support the conclusions?

Reviewer #1: Yes

Reviewer #2: Yes

2. Has the statistical analysis been performed appropriately and rigorously? 

Reviewer #1: Yes

Reviewer #2: Yes

3. Have the authors made all data underlying the findings in their manuscript fully available?

Reviewer #1: Yes

Reviewer #2: Yes

4. Is the manuscript presented in an intelligible fashion and written in standard English?

Reviewer #1: Yes

Reviewer #2: Yes

5. Review Comments to the Author

Reviewer #1: Major comments

P5

The exclusion criteria include cases that have undergone intraocular surgery, but since the analysis includes surgical cases, should the exclusion criteria be changed?

p7

The fundus photograph is not a pure plane. In the examination images or images displayed on the viewer, was the calculation of the ERM area performed taking into account that the fundus photograph is not a plane but closer to a sphere? If not, I think this should be added to the limitations.

P11

The New Aniseikonia Test (NAT) is a test that quantitatively evaluates the relative aniseikonia symptoms of both eyes, and if the other eye is not normal, the test results are considered meaningless.

Cases with ERM in both eyes or cases with ERM in one eye but with a macular disease that may cause aniseikonia in the other eye should be excluded because it may not be possible to evaluate aniseikonia caused solely by ERM.

P13

The cases that underwent surgery were classified into the progression group, with 23 eyes in this group. It is stated that surgery was performed on 9 eyes. This means that 14 eyes in the progression group did not undergo surgery. However, at the 24-month mark in Fig. 1, the number of cases is 19, and the cases that underwent surgery are also included in the analysis of area changes.

In that case, the effects of changes in visual acuity and area due to surgery would be confused with changes in visual acuity and area due to the natural course of ERM, and we think that cases that underwent surgery should be excluded from the analysis of changes in area and visual acuity.

Minor comments

P6

In the following sentence, “The lens status was determined by slit-lamp examination and graded according to the LOCS III system.15,” the citation number is not included in parentheses. Please revise this.

P14 The P-values below should be either P<0.05 or corrected to “=” as appropriate.

P14

P<0.029, P<0.041, P<0.037, P<0.031,

P15

P<0.019,

P16

P<0.036, P<0.016

Reviewer #2: This is a well-written paper investigating the relationship between BCVA and ERM area change.

The authors described that ERM area reduction significantly correlates with vision loss in patients

with iERM with good vision.

Factors affecting visual acuity in ERM cases include the presence or absence of central foveal depression, presence or absence of EIFL, EZ/ELM continuity, central foveal deviation, and central foveal retinal thickness. In light of these points, the following points should be considered in this report.

1.

Although this study included stage 1 or 2 cases, was there no change in the final ERM? Did it progress?

In other words, is it confirmed that EIFL did not affect vision during the course of the study?

2.

Although retinal thickness at various sites is evaluated, shouldn't the continuity of EZ and ELM be the subject of evaluation rather than that?

It may be necessary to consider the fact that the continuity of the EZ and ELM was lost in the progression group.

The results indicate that vision loss occurs when the overall contraction of the ERM is strong, but does this mean that there is no continuity of the EZ or ELM in cases with strong contraction, or that there is eccentricity in the FAZ?

The continuity of EZ and ELM should be discussed.

3.

In Figure 2, the intersection of the segmentation lines in both cases is at the same location on the fundus at all time points.

Did the position of the central fossa change during the course of the disease due to ERM traction?

4.

As mentioned in the limitaion, there is little consideration of cataracts.

If it means that there is no effect of cataract, the results of LOCS during the course of the disease should be described.

Also, in the progression group, did the patients who underwent vitrectomy recover their vision in the early postoperative period?

If cataract surgery was performed at the same time as the cataract surgery and vision recovered in the early postoperative period, could this be considered a cataract-related vision loss?



I have the impression that the same information is stated many times in the discussion section. It would be easier to read if it was summarized and presented.

6. PLOS authors have the option to publish the peer review history of their article (what does this mean? ). If published, this will include your full peer review and any attached files.

**Do you want your identity to be public for this peer review?** For information about this choice, including consent withdrawal, please see our Privacy Policy .

Reviewer #1: No

Reviewer #2: No

---

## [Author Response · Author response to Decision Letter 1]

6 Aug 2025

Point-by-point response to the reviewer’s comments

Dear Editor & Reviewers,

We appreciate all suggestions and corrections. All authors have carefully reviewed these comments. Our detailed responses to comments are addressed below.

Review Comments & Answers

Reviewer #1: Major comments

1. p5

The exclusion criteria include cases that have undergone intraocular surgery, but since the analysis includes surgical cases, should the exclusion criteria be changed?

Response: The original intention was to exclude ocular surgeries performed prior to enrollment; however, the criterion appears to have been inaccurately described. We appreciate your valuable feedback. The pointed out parts have been corrected as follows. “(6) prior ocular surgery other than cataract surgery.” (Last sentence of 3rd paragraph on Page 5).

2. p7

The fundus photograph is not a pure plane. In the examination images or images displayed on the viewer, was the calculation of the ERM area performed taking into account that the fundus photograph is not a plane but closer to a sphere? If not, I think this should be added to the limitations.

Response: In this study, we measured the length and area of the ERM within a 5-mm zone. Although fundus photography is a two-dimensional imaging modality, the discrepancy between the actual tissue length and the measured distance on the image is expected to be similar to that observed in OCT-based measurements. In general OCT studies, the retinal curvature in the macular region is considered sufficiently flat within the narrow field of view, and therefore, it is unlikely to have significantly affected the outcomes of our study. In our measurements on fundus photographs, the chord lengths were 5.0, 4.0, 3.0, 2.0, 1.0, 0.5, and 0.25 mm, corresponding to arc lengths of 5.036 mm, 4.019 mm, 3.008 mm, 2.002 mm, 1.000 mm, 0.500 mm, and 0.250 mm, respectively. The arc-to-chord ratios for these distances were 1.007, 1.005, 1.003, 1.001, 1.000, 1.000, and 1.000. For chord lengths below 3 mm, the corresponding central angles are less than 15°, for which the small-angle approximation is mathematically applicable, justifying the use of a flat-surface assumption. Such small numerical differences are unlikely to produce meaningful errors even when squared for area measurements. Since our measurements were performed within a 5-mm zone, assuming a flat surface in the analysis of fundus photographs is not expected to have introduced any significant inaccuracies. In addition, to minimize errors caused by severe retinal curvature, eyes with high myopia were excluded based on our exclusion criteria.

3. P11

The New Aniseikonia Test (NAT) is a test that quantitatively evaluates the relative aniseikonia symptoms of both eyes, and if the other eye is not normal, the test results are considered meaningless.

Cases with ERM in both eyes or cases with ERM in one eye but with a macular disease that may cause aniseikonia in the other eye should be excluded because it may not be possible to evaluate aniseikonia caused solely by ERM.

Response: As reviewer correctly pointed out, aniseikonia should not be measured in cases with bilateral ERM. We have added the sentence "Aniseikonia measurements were not performed in cases with bilateral ERM." at the end of the fourth paragraph on page 6. In addition, the related values have been revised accordingly; however, these changes did not affect the statistical significance of the results. No macular abnormalities were found in the fellow eyes.

4. P13

The cases that underwent surgery were classified into the progression group, with 23 eyes in this group. It is stated that surgery was performed on 9 eyes. This means that 14 eyes in the progression group did not undergo surgery. However, at the 24-month mark in Fig. 1, the number of cases is 19, and the cases that underwent surgery are also included in the analysis of area changes.

In that case, the effects of changes in visual acuity and area due to surgery would be confused with changes in visual acuity and area due to the natural course of ERM, and we think that cases that underwent surgery should be excluded from the analysis of changes in area and visual acuity.

Response: This study aimed to investigate the clinical characteristics of patients who initially presented with good visual acuity, reflecting recent trends in retinal surgery in which procedures are performed even at relatively good visual acuity levels. Accordingly, we enrolled only patients with a baseline visual acuity of 20/25 or better. However, once a patient experienced a loss of two or more lines of vision, their visual acuity would fall below 20/32, which meets the surgical indication. As a result, no patients in the progression group remained untreated once this level of visual decline occurred. To avoid losing data from cases that might have undergone surgery too early, we included only patients whose disease progression could be followed for at least 12 months. For those who underwent surgery after more than 12 months of follow-up, the last available data point prior to surgery was used in the analysis, as stated in the Methods section. For analyses investigating the relationship between visual acuity and other factors, we used either the last follow-up data or the 24-month data to evaluate the impact of visual decline. However, to assess the trends in changes in visual acuity and ERM area over time, we included data from all patients at each time point according to their follow-up duration. As shown in Table 1, all included patients had at least 24 months of follow-up. Among the 9 eyes that underwent surgery, 4 underwent surgery between 12 and 24 months in the progression group. The remaining 5 eyes underwent surgery after additional follow-up due to visual decline. Therefore, 19 eyes were included in the 24-month analysis.

Due to a typographical error, this was previously stated as "within 24 months" and has now been corrected in the Results section as follows:

“During the follow-up period, ERM surgery was performed in 9 out of 46 eyes (19.6%) with a median BCVA of 0.30 (Snellen acuity 20/40, interquartile range: 0.30–0.30), all of which belonged to the progression group. Four of these eyes underwent surgery within 24 months.”

Minor comments

5. P6

In the following sentence, “The lens status was determined by slit-lamp examination and graded according to the LOCS III system.15,” the citation number is not included in parentheses. Please revise this.

Response: As you suggested, we have revised the sentence as follows: “The lens status was determined by slit-lamp examination and graded according to the LOCS III system [15].” (Last sentence of 3rd paragraph on Page 6).

6. P14 The P-values below should be either P<0.05 or corrected to “=” as appropriate.

P14

P<0.029, P<0.041, P<0.037, P<0.031,

P15

P<0.019,

P16

P<0.036, P<0.016

Response: In accordance with the comment, the following P-values were revised to P < 0.05: P < 0.029, P < 0.041, P < 0.037, and P < 0.031 on page 14; P < 0.019 on page 15; and P < 0.036 and P < 0.016 on page 16.

Reviewer #2: This is a well-written paper investigating the relationship between BCVA and ERM area change.

The authors described that ERM area reduction significantly correlates with vision loss in patients

with iERM with good vision.

Factors affecting visual acuity in ERM cases include the presence or absence of central foveal depression, presence or absence of EIFL, EZ/ELM continuity, central foveal deviation, and central foveal retinal thickness. In light of these points, the following points should be considered in this report.

1. Although this study included stage 1 or 2 cases, was there no change in the final ERM? Did it progress?

In other words, is it confirmed that EIFL did not affect vision during the course of the study?

Response: The mean OCT stage in the stable group was 1.6 ± 0.5 at baseline, and no eyes showed any change in stage during the follow-up period. In contrast, in the progression group, the mean OCT stage significantly increased from 1.7 ± 0.4 at baseline to 1.9 ± 0.5 at the final follow-up (P=0.014, Wilcoxon’s signed rank test). Specifically, three eyes progressed from stage 1 to 2, and another three eyes progressed from stage 2 to 3. Among the three eyes that progressed to stage 3, the presence of EIFL did not affect the two-line loss of visual acuity (P=0.233, Fisher’s exact test). However, due to the small number of eyes with EIFL, the analysis may have been underpowered to demonstrate a statistically significant association. As a result, the relationship between EIFL and visual acuity was not described in the manuscript. These results were added to the Results section as follows:

“In the stable group, no eyes showed a change in OCT stage during the follow-up period. However, in the progression group, the OCT stage significantly increased from a mean of 1.7 ± 0.4 at baseline to 1.9 ± 0.5 at the final follow-up (P=0.014). Specifically, stage progression was observed in 3 eyes from stage 1 to 2 and in another 3 eyes from stage 2 to 3.”

(Fourth paragraph on page 16). Additionally, OCT stage was included in Table 1 as part of the baseline characteristics.

2. Although retinal thickness at various sites is evaluated, shouldn't the continuity of EZ and ELM be the subject of evaluation rather than that?

It may be necessary to consider the fact that the continuity of the EZ and ELM was lost in the progression group.

The results indicate that vision loss occurs when the overall contraction of the ERM is strong, but does this mean that there is no continuity of the EZ or ELM in cases with strong contraction, or that there is eccentricity in the FAZ?

The continuity of EZ and ELM should be discussed.

Response: Quantitative analysis of ERM changes using OCT was challenging due to variations in morphology depending on the cross-sectional location and differences in retinal thickness depending on the shape of the ERM. To address this limitation, we evaluated the impact of ERM area changes on visual acuity as the disease progressed.

Because this study focused on a very early stage of the disease, the amount of outer retinal thickness change was relatively small (see S1 Table). Although the differences in changes between inner and outer retinal thickness did not reach statistical significance and were therefore not included in the main text, the P value was approximately 0.067—close to the threshold—suggesting that changes in inner retinal thickness may be a key factor contributing to visual acuity decline. Furthermore, although not stated in the manuscript, no eyes exhibited disruption of the EZ or ELM from baseline to the end of follow-up.

For this reason, we thought that it is better not to include a broad range of OCT parameters such as EZ and ELM continuity, as doing so would have made the manuscript overly extensive and potentially distracted from the main focus.

3. In Figure 2, the intersection of the segmentation lines in both cases is at the same location on the fundus at all time points.

Did the position of the central fossa change during the course of the disease due to ERM traction?

Response: As you mentioned, foveal displacement of the inner retina may occur due to ERM contraction. However, assessing the exact shift of the foveal center requires a fixed anatomical reference such as the choroid, which does not align with the methodology of our study; therefore, this aspect was not evaluated. In this study, the central region was determined by referencing the position of the foveal bulge on OCT. Accordingly, the intersection lines in Figure 2 are located at the same anatomical position. Using the inner retina as a reference may be advantageous when assessing quadrant-specific changes of the membrane itself. However, this approach has limitations in analyzing correlations with retinal thickness changes. In our previous study, postoperative foveal displacement in the inner retina following ERM surgery was approximately 0.2 mm. Additionally, displacement caused by ILM peeling in macular hole cases was about 0.14 mm, and the difference between fellow eyes was 0.028 mm. Considering these findings, we believe that the subtle displacement of the fovea would not have significantly affected the outcomes of the present study.

We have added the following sentence to the Methods section:

“The macular center was determined by referencing the vertex of the foveal bulge on OCT.”

(Last sentence in the third paragraph on page 7.)

Reference

(1) Lee SM, Park KH, Kwon HJ, Park SW, Byon IS, Lee JE. Displacement of the Foveal Retinal Layers After Macular Hole Surgery Assessed Using En Face Optical Coherence Tomography Images. Ophthalmic Surg Lasers Imaging Retina. 2019;50(7):414-422.

(2) Lee SM, Park SW, Byon I. Topographic changes in macula and its association with visual outcomes in idiopathic epiretinal membrane surgery. PLoS One. 2025;20(1):e0316847. Published 2025 Jan 9.

4. As mentioned in the limitaion, there is little consideration of cataracts.

If it means that there is no effect of cataract, the results of LOCS during the course of the disease should be described.

Also, in the progression group, did the patients who underwent vitrectomy recover their vision in the early postoperative period?

If cataract surgery was performed at the same time as the cataract surgery and vision recovered in the early postoperative period, could this be considered a cataract-related vision loss?

Response: As suggested, we evaluated changes in lens status and their association with vision loss.

According to the LOCS grading system, the nuclear opalescence (NO) and cortical cataract (C) scores in the stable group were 2.0 ± 0.0 and 0.9 ± 0.7 at baseline, and 2.2 ± 0.4 and 0.9 ± 0.7 at the final follow-up, respectively. In the progression group, the baseline scores were 1.9 ± 0.2 for NO and 0.6 ± 0.7 for C, which changed to 2.0 ± 0.4 and 0.6 ± 0.7 at the final follow-up, respectively. There was no statistically significant increase in cataract severity in either group (stable group: P = 0.083 for NO, P = 0.317 for C; progression group: P = 0.157 for NO, P = 1.000 for C; Wilcoxon’s signed rank test).

Additionally, there was no statistically significant association between cataract severity and vision loss (baseline: P = 0.335 for NO, P = 0.561 for C; follow-up: P = 0.126 for NO, P = 0.455 for C; Chi-square analysis).

Due to ILM peeling, early postoperative visual improvement was not observed within 1–2 months. While we cannot completely rule out the influence of cataract-related visual changes, only one group experienced visual deterioration despite both groups having similar levels of cataract.

Considering this and previous findings noted in the limitations section, it is unlikely that cataract progression had a significant impact on the visual outcomes of this study.

Accordingly, the following sentence was added to the end of the baseline characteristics section in the Results (page 12):

“In the stable group, nuclear opalescence and cortical cataract scores were 2.0 ± 0.0 and 0.9 ± 0.7 at baseline, and 2.2 ± 0.4 and 0.9 ± 0.7 at the final follow-up, respectively. In the progression group, the scores were 1.9 ± 0.2 and 0.6 ± 0.7 at baseline, and 2.0 ± 0.4 and 0.6 ± 0.7 at the final follow-up, respectively. There was no significant progression of cataract in either group.”

. I have the impression that the same information is stated many times in the discussion section. It would be easier to read if it was summarized and presented.

Response: Following the reviewer’s suggestions, we carefully revised the Discussion section to minimize redundancy and omit content deemed unnecessary.

---

## [Decision Letter · Decision Letter 1]

17 Aug 2025

Idiopathic Epiretinal Membrane Area Changes in Eyes with Good Vision and the Association with Visual Function

PONE-D-25-36863R1

Dear Dr. Lee,

We’re pleased to inform you that your manuscript has been judged scientifically suitable for publication and will be formally accepted for publication once it meets all outstanding technical requirements.

Kind regards,

Daisuke Nagasato

Academic Editor

PLOS ONE

Additional Editor Comments (optional):

Reviewers' comments:

Reviewer's Responses to Questions

**Comments to the Author**

1. If the authors have adequately addressed your comments raised in a previous round of review and you feel that this manuscript is now acceptable for publication, you may indicate that here to bypass the “Comments to the Author” section, enter your conflict of interest statement in the “Confidential to Editor” section, and submit your "Accept" recommendation.

Reviewer #1: All comments have been addressed

Reviewer #2: All comments have been addressed

2. Is the manuscript technically sound, and do the data support the conclusions?

Reviewer #1: Yes

Reviewer #2: Yes

3. Has the statistical analysis been performed appropriately and rigorously? 

Reviewer #1: Yes

Reviewer #2: Yes

4. Have the authors made all data underlying the findings in their manuscript fully available?

Reviewer #1: Yes

Reviewer #2: Yes

5. Is the manuscript presented in an intelligible fashion and written in standard English?

Reviewer #1: Yes

Reviewer #2: Yes

6. Review Comments to the Author

Reviewer #1: (No Response)

Reviewer #2: Thank you for your responses and revisions to my comments.

The authors were fully responsive to the reviewer's questions and requests for corrections.

The manuscript has been fully revised and appears good.

7. PLOS authors have the option to publish the peer review history of their article (what does this mean? ). If published, this will include your full peer review and any attached files.

**Do you want your identity to be public for this peer review?** For information about this choice, including consent withdrawal, please see our Privacy Policy .

Reviewer #1: No

Reviewer #2: No

---

## [Editor Report · Acceptance letter]

PONE-D-25-36863R1

PLOS ONE

Dear Dr. Lee,

I'm pleased to inform you that your manuscript has been deemed suitable for publication in PLOS ONE. Congratulations! Your manuscript is now being handed over to our production team.

Kind regards,

on behalf of

Dr. Daisuke Nagasato

Academic Editor

PLOS ONE